# ARISE—The Accuracy Evaluation of a Patient-Specific 3D-Printed Biopsy System Based on MRI Data: A Cadaveric Study

**DOI:** 10.3390/bioengineering11101013

**Published:** 2024-10-11

**Authors:** Robert Möbius, Dirk Winkler, Fabian Kropla, Marcel Müller, Sebastian Scholz, Erdem Güresir, Ronny Grunert

**Affiliations:** 1Faculty of Medicine, University Clinic of Leipzig, 04103 Leipzig, Germany; moebius@biosaxony.com (R.M.); dirk.winkler@medizin.uni-leipzig.de (D.W.); fabian.kropla@medizin.uni-leipzig.de (F.K.); erdem.gueresir@medizin.uni-leipzig.de (E.G.); 2Fraunhofer Institute for Machine Tools and Forming Technology, 02763 Zittau, Germany; marcel.mueller@iwu.fraunhofer.de (M.M.); sebastian.scholz@iwu.fraunhofer.de (S.S.)

**Keywords:** stereotaxy, cerebral biopsy, intracranial lesion, 3D printing

## Abstract

Background: Brain biopsy is required for the accurate specification and further diagnosis of intracranial findings. The conventional stereotactic frames are used clinically for biopsies and offer the highest possible precision. Unfortunately, they come with some insurmountable technical and logistical limitations. The aim of the present work is to determine the clinical precision in the needle biopsy of the human brain using a new patient-specific stereotactic navigation device based on 3D printing. Methods: MRI data sets of human cadaver heads were used to plan 32 intracranial virtual biopsy targets located in different brain regions. Based on these data, 16 individualized stereotactic frames were 3D-printed. After the intraoperative application of the stereotactic device to the cadaver’s head, the actual needle position was verified by postoperative CT. Results: Thirty-two brain areas were successfully biopsied. The target point accuracy was 1.05 ± 0.63 mm, which represents the difference between the planned and real target points. The largest target point deviation was in the coronal plane at 0.60 mm; the smallest was in the transverse plane (0.45 mm). Conclusions: Three-dimensional-printed, personalized stereotactic frames or platforms are an alternative to the commonly used frame-based and frameless stereotactic systems. They are particularly advantageous in terms of accuracy, reduced medical imaging, and significantly simplified intraoperative handling.

## 1. Introduction

The removal of pathological tissue with the use of brain biopsy is an essential prerequisite for the clarification of intracranial findings using histopathological examination and for the determination of future therapeutic procedures resulting from this. In order to reach the desired target area and to minimize trauma to the surrounding parenchymal tissue, high precision in tissue sampling and a minimally invasive biopsy procedure are required [1].

Image-guided (MRI and CT) stereotactic systems, such as conventional stereotactic frames or optically navigated structures, have been clinically established for this purpose. All of these require complex imaging and are very time-consuming and expensive in terms of acquisition, maintenance, calibration, and storage, and complex handling requires in-depth knowledge and a large learning curve. Frame-based stereotaxy remains the “gold standard” for brain biopsy [2,3,4,5,6,7,8,9]. The conventional methods of frame stereotaxy that are commonly used today are opposed to the demands from neurosurgeons for a simple and inexpensive system with comparable accuracy. In addition, the aim is to eliminate the main disadvantages of the current frame-based stereotaxic systems, such as their complicated handling, heaviness, and the lack of patient comfort. It is also important to eliminate the fusion of MR data with additional CT data, which is often required for conventional frame stereotaxy.

An increase in additively manufactured single-use products has recently been observed in the medical technology market. These combine a high degree of individualization through patient-specific production with simplicity and the quick availability of the device. The other advantages include the use of medical-grade plastic materials that are lightweight and biocompatible. In a previous research project, a patient-specific stereotaxic device for intracranial biopsy was developed specifically for the veterinary sector [10,11]. The system was then further developed and transferred to the field of human neurosurgery. The aim of the present work was the discussion and evaluation of an MRI-based, patient-specific, stereotactic, 3D-printed device for intracranial biopsy. The basis for this was the achieved precision of the stereotactic instrument (biopsy needle), which was secured and documented by CT. The discussion regarding the possible influences on stereotaxic accuracy with regard to the depth of the biopsied process is interesting.

## 2. Material and Methods

### 2.1. Body Donation

Body acquisition and institutional approval was managed by the Institute of Anatomy (Leipzig, Germany), being a part of the body donor program regulated by the Saxonian Death and Funeral Act of 1994 (third section, paragraph 18 item 8). One cadaveric ethanol-fixed head was used for this purpose. The head tissue has no external injuries or scars of recent neurosurgical procedures. The baseline data of the body donor are shown in Table 1.

### 2.2. Preparation

All the preparation steps were performed by an experienced neurosurgeon (senior physician level). Skin incisions were made on the donated head, and ten titanium-made screw bone anchors (type 5 mm waypoint, FHC Inc., Bowdoin, ME, USA) were applied into the cortical bone of the cranial dome. Calculated access corresponds to the shortest distance between the skull surface and the target point and defines the placement of the bone anchors. The trajectory was plotted inside the center of the placed bone anchors.

Three anchors were placed laterally on both sides. Four anchors were placed behind each other on the medial axis. The minimum distance between each anchor was 30 mm. A presentation of the bone anchor configuration is shown in Figure 1.

Two markers visible on MRI and CT were used. These markers contain of a cylindric body and two axially arranged gel balls inside. Ten MRI markers were screwed into each anchor. The marker configuration is shown in Figure 2.

### 2.3. Clinical Imaging

#### 2.3.1. MR Imaging

Diagnostic imaging was performed in the Department of Radiology (University of Leipzig). T1-weighted MR scans (slice thickness 1.0 mm) of the cadaveric head with attached MR markers were obtained using 3-Tesla MRI and an 18-channel matrix head coil (Achieva dStream, Philips Medical Systems, Hamburg, Germany).

#### 2.3.2. CT Imaging

Additionally, the same areas were registered using helix CT with a 0.67 mm slice thickness (Ingenuity Core, Philips Medical Systems, Hamburg, Germany), and then again as the intraoperative control.

CT scanning was only necessary as a measurement tool to determine the accuracy of the patient-individualized stereotactical device as the biopsy needle cannot be scanned due to the metal with MRI. For future surgical applications, only thin-layered MR imaging is required.

### 2.4. Virtual Target Planning

The MRI DICOM data set was imported into FDA-approved software D2P^®^ (Ver. 1.02.2055) (DICOM to Print, 3D Systems Inc., Rock Hill, SC, USA). Target point planning was performed by an experienced neurosurgeon (senior physician level). Thirty-two biopsy target points and associated skull bone entry points were defined and marked on transversal T1-weighted MR images in the brain data set. The targets were alternated, localized in the left and right brain hemispheres. The targets were also distributed with respect to their depth in the brain (from <30 to >50 mm). For the resulting 32 target trajectories, three surrounding bone anchor points for the later mounting of the three-legged navigation devices were determined by a supervising engineer.

### 2.5. Three-Dimensional Printing

Based on the given image data, 16 navigation devices, each consisting of a target of the respective right and left brain hemispheres, were defined. All the geometric data of the anchorage, target, and skull entry points were exported as Standard Triangle Language (STL) files from the MR image data sets. The software GOMInspect 2019 (GOM GmbH, Braunschweig, Germany) was used to read out and export the numeric values of the coordinate system.

In terms of reasonable geometric constellations, the stereotaxic platforms were designed by using SolidWorks Professional 2019 (Dassault Systèmes SolidWorks Corp., Vélizy-Villacoublay France). Each stereotactic platform consists of a three-legged frame secured to the defined cranial bone anchors and biopsy ports for two trajectories. The design is shown in Figure 3. In addition, a length template was constructed, and with the aid of this, the subsequent needle length for the biopsy was set using submillimeters. Each biopsy port and length template was labeled with a study number.

The 16 devices and 32 length templates were manufactured from polyamide (type: PA 12) by using the 3D-printing technique MultiJetfusion 580 color (Hewlett-Packard Inc., Palo Alto, CA, USA). MJF technology is a powder bed process. PA12 in powder form is applied layer by layer in the building platform. After each individual layer, print heads are used to apply an agent in the areas where the 3D model is to be produced. After this application, a thermal radiator moves over the current layer and heats it. The thermal energy is absorbed in particular by the powder particles sprayed with the agent, which then fuse together. This process is repeated for each additional layer.

After the printing process, the surface of the devices and templates were glass bead-blasted with an SMG 50 Rapid Ex machine (MHG GmbH, Düsseldorf, Germany).

### 2.6. Intraoperative Implementation

The donor head was placed on a surgical table, and the individual stereotactic frame was secured to the three associated bone anchors using M3 screws. A 10 mm incision was made into the skin until the skull surface was exposed. A borehole of 3 mm in diameter was drilled by using a rose burr with head diameter of 3 mm (Electric Pen Drive, DePuy Synthes, West Chester, PA, USA) after placing a drill sleeve on the tool guide.

The dura mater was perforated using a 0.3 mm hypodermic needle. A spacer was secured to the titanium-made 2.5 mm Sedan side-cutting biopsy needle (ELEKTA, Stockholm, Sweden) in order to restrict the needle advancement to the desired depth (Figure 4).

The procedure described was repeated in the same way for all the 32 trajectories.

### 2.7. Accuracy Evaluation

To verify the target points, postoperative control CT with a slice thickness of 0.67 mm was conducted after the positioning of the biopsy needle. Subsequently, the cranial calotte and the biopsy needle were segmented from the CT image data using Image Software D2P^®^ (Ver. 1.02.2055) again.

With help of the preoperatively acquired CT, the MRI target points were merged into one coordinate system by using the MR contrast markers (visible in MRI and CT). The CT data sets (preoperative and postoperative) were fused with the skull bone data into one coordinate system. Geometric regression was used to compare the planned virtual target point and the real target point reached on CT (biopsy needle tip) (Figure 5). The Euclidian deviation of the two target points as well as the segmented biopsy needle diameter were determined.

### 2.8. Statistical Analyses

Data processing and statistical comparison was performed by using Microsoft Excel 2013 (Redmond, WA, USA) and SPSS 24.0 IBM Inc. (Armonk, NY, USA). For evaluation, mean values, medians, and standard deviations were calculated. Statistical comparisons were made by using a non-parametric Mann–Whitney U test. Correlation was proofed with a Spearman Rho test. The significance level was set to 0.05.

## 3. Results

Based on T1-weighted MRI data set of the cadaveric head, 32 intracranial target points and trajectories were determined. Using the MJF 3D-printing technique, the navigation devices were manufactured from polyamide material (PA12). All the target points were biopsied in the cadaveric brain with a postoperative CT control. All the targets were evaluated by the registration of additional CT scans as a measurement tool (preoperative and postoperative).

### 3.1. Accuracy Target Points

The mean deviation in the preoperatively planned and real biopsy target points was 1.05 ± 0.63 mm, which ranged from 0.24 mm to 2.52 mm, and the median was 0.85 mm. The deviation in the x-plane (sagittal) was 0.56 ± 0.53 mm, in the y-plane (coronal), it was 0.60 ± 0.53 mm, and in the z-plane (transversal), it was 0.45 ± 0.24 mm. The titanium biopsy needle used for this study had a mean diameter of 2.89 ± 0.23 mm. The intracranial needle depth ranged from 22.41 mm to 72.39 mm (mean: 41.36 ± 12.51 mm). There was no linear correlation between the target point deviation and the lesion depth (CCSpearman = 0.214).

### 3.2. Brain Hemispheres and Lesion Depth

The mean values regarding the brain hemispheres differed at the non-significant level (*p* = 0.792), with that of the right hemisphere being 0.97 ± 0.53 mm (Median_rigth_ = 0.85 mm) and that of the left hemisphere being 1.14 ± 0.72 mm (Median_left_ = 0.87 mm).

The influence of lesion depth in the brain is also not significant (*p* = 0.954) with values of 0.99 ± 0.49 mm (Median_superficial_ = 0.86 mm, less than 30 mm below the cranial bone) and 1.14 ± 0.80 mm (Median_deep_ = 0.73 mm, more than 30 mm below the cranial bone).

### 3.3. Manufacturing Parameters

The mean weight of the PA12 manufactured stereotactical devices was only 62.4 ± 6.0 g, ranging from 50.8 g to 73.6 g. Four navigation devices were manufactured in one 3D building job. The manufacturing times of the individual 3D printing jobs ranged between 450 and 580 min. There were no production rejects while manufacturing when using the MJF 3D technique. All the results are shown in Table 2.

## 4. Discussion

The objective of this study was to prove the clinical precision in human brain biopsy of a new developed patient individual stereotactic device. Based on the MRI data of a cadaveric head, the target points and the trajectories were determined. Using the MJF 3D-printing technique, the stereotactic devices were manufactured from polyamide material and used for CT-controlled biopsy [12,13,14]. Three-dimensional printing contributes to cost reduction because no large investment for conventional stereotactic frames or surgical navigation systems is required. The material used (PA 12) has been approved and certified for steam sterilization (134 °C) and can be used on patients in an OR environment without hesitation.

There were no problems with the additive manufacturing and handling of the 3D-printed components. All the devices could be mounted precisely onto the registered skull bone anchors and used for craniotomy and biopsy needle placement. To the authors’ knowledge, this is the first procedure conducted for intracranial navigation exclusively based on MRI planning without CT-specified reference markers.

The patient-individual stereotactic devices allowed for needle placement within a mean accuracy of about 1 mm deviation (1.05 ± 0.63 mm) of the intended target in all the procedures. The maximum deviation was about 2.5 mm, which is still tolerable in the field of stereotactical interventions. The results show further small differences in terms of the brain hemisphere and the lesion depth, but at the non-significant level.

The mean values of the results show comparable accuracy to those of other frameless systems like the Vertek Aiming Device (Medtronic Inc., Louisville, USA) [15,16]. Other frameless systems like VarioGuide (BrainLAB AG, Feldkirchen, Germany) report an accuracy that ranges from 2.5 mm to 8.3 mm [17,18,19].

Bjartmarz et al. compared the accuracy between the conventional stereotactic and frameless systems for deep brain stimulation. Their results were 1.2 ± 0.6 for conventional and 2.5 ± 1.4 mm for frameless and systems [20]. The frameless VarioGuide and the frame-based stereotactic system were analyzed by Bradac et al., with results of 2.7 ± 1.1 mm for the conventional system and 2.9 ± 1.3 mm for the frameless VarioGuide [21]. Hodge et al. evaluated a frameless system for deep brain stimulation, with an accuracy of 1.79 ± 1.02 mm [22].

Our results, with an accuracy of 1.05 ± 0.63 mm, are comparable to those of the conventional frame-based stereotactic systems in comparison to the mentioned studies.

Compared to the frame-based stereotactical systems, there are many advantages in terms of intraoperative handling, time efficiency, and patient safety. This aspect is also described by other authors, as they report no significant differences exist between frame-based and frameless biopsy in diagnostic yield, morbidity, or mortality. Frameless biopsy is associated with shorter procedural times relative to those of frame-based biopsy [9,23,24]. The operating time is approx. 2 h for the conventional systems. This time is also preceded by a CT scan under general anesthesia, trajectory determination, and the calculation and assembly of the stereotactic frame and axis adjustment. This compares with an operating time of just 25 min with the new 3D-printed system, including the new workflow.

Instead of a CT scan, only thin-layered MRI is required for the planning and construction of the new stereotactic device. Thus, the patient is not exposed to potential harmful X-rays. Furthermore, no image fusion for CT and MRI has to be performed. Therefore, an image fusion error can be avoided. If there thin-layered MRI is not available, the CT imaging data could be applied to acquire the images for the construction process.

The 3D-printed system has great material advantages with regards to sterilizability and medical use characteristics, which are fully documented in the material certificates and data sheets. The other benefits are the light-weight construction (from 60 to 80 g) compared to 2000–3000 g for the conventional stereotactic frames and the manufacturing costs. The patient-specific 3D-printed stereotactic device eliminates all the resterilization and reprocessing costs, as well as the maintenance, recalibration, repair, and storage costs.

## 5. Conclusions

The developed patient-specific 3D-printed system provides neurosurgeons with an option for precise and easy-to-handle stereotactic surgery to place biopsy needles. The possible errors due to the axis adjustment of the conventional stereotactic systems are avoided, as all the spatial axes are implemented in the 3D-printed system, and no adjustments to the distances and angles are required. This makes the stereotactic principle available to non-experienced neurosurgeons.

A future application could be the placement of electrodes for deep brain stimulation.

## 6. Limitations

The results in the given study refer to only one body donor head model. The body donor tissue was ethanol-fixed according to the house protocol from the Institute of Anatomy (Leipzig University) with all the associated post-mortem changes in tissue properties. The precision of needle segmentation was dependent on the quality of the intraoperative CT scan (scatter radiation, etc.). The maximum precision to be achieved is largely limited to the MRI planning procedure, with a minimum slice thickness of 1.0 mm.

## Figures and Tables

**Figure 1 bioengineering-11-01013-f001:**
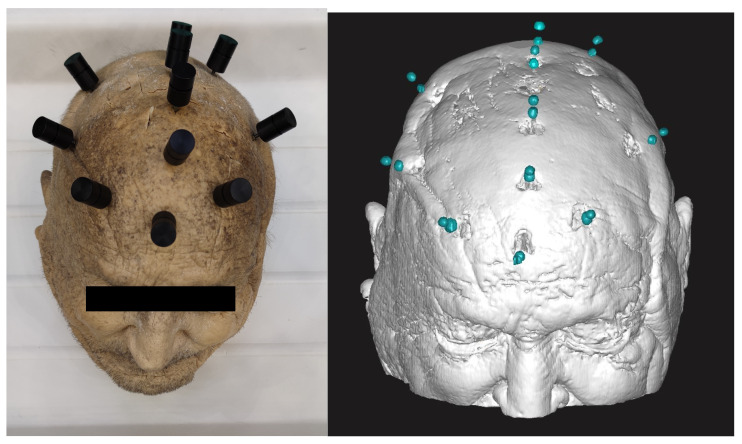
The placement of MRI markers on the body donor head and an MRI scan.

**Figure 2 bioengineering-11-01013-f002:**
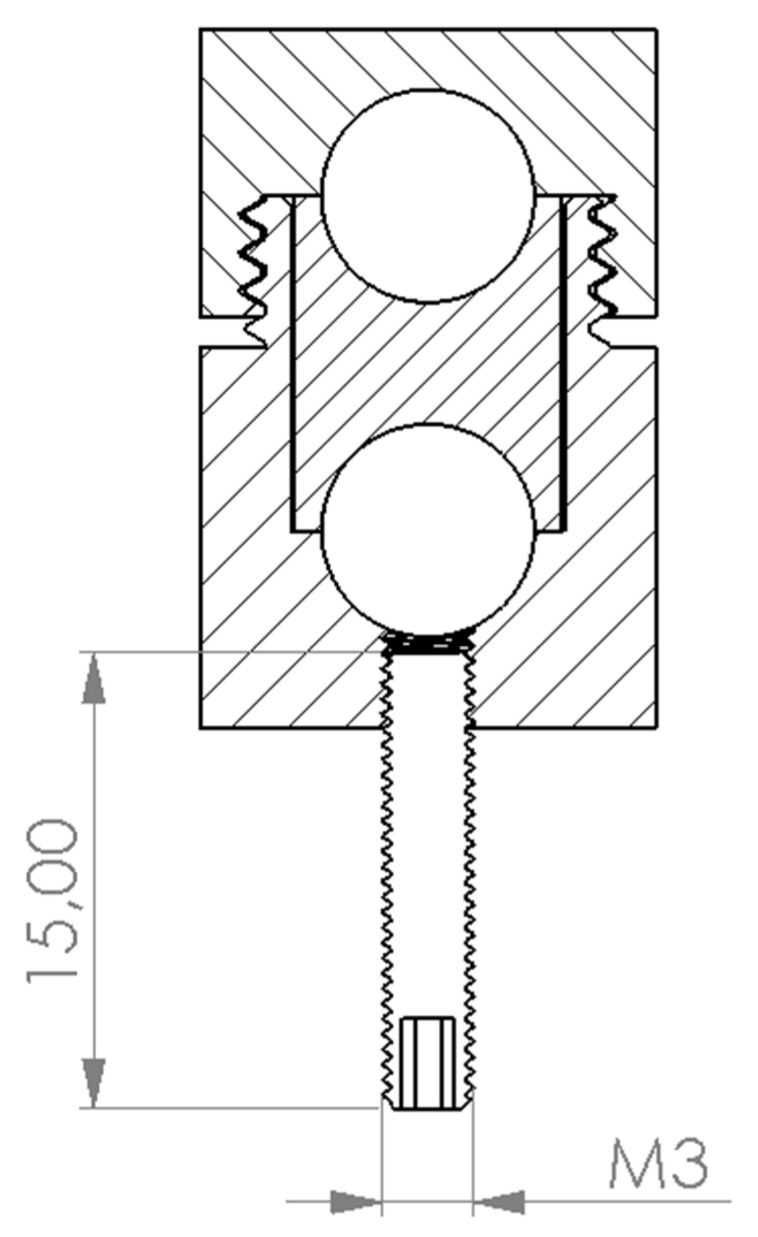
MRI marker configuration.

**Figure 3 bioengineering-11-01013-f003:**
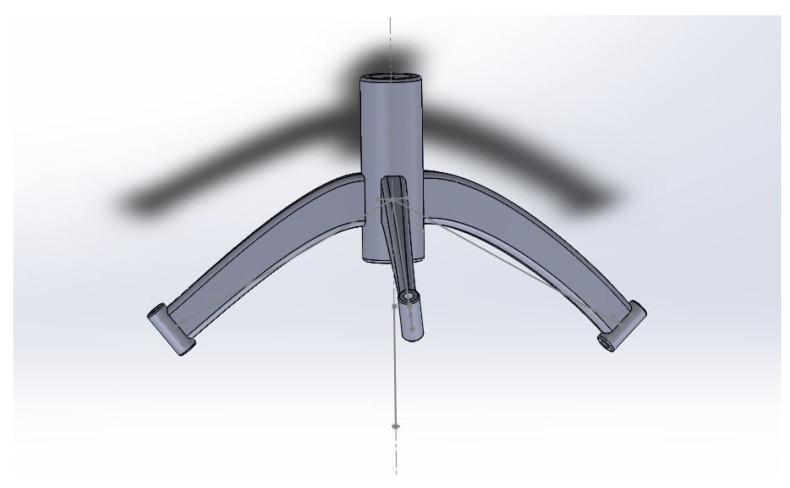
Design of patient-individual stereotactical device for intracranial biopsy.

**Figure 4 bioengineering-11-01013-f004:**
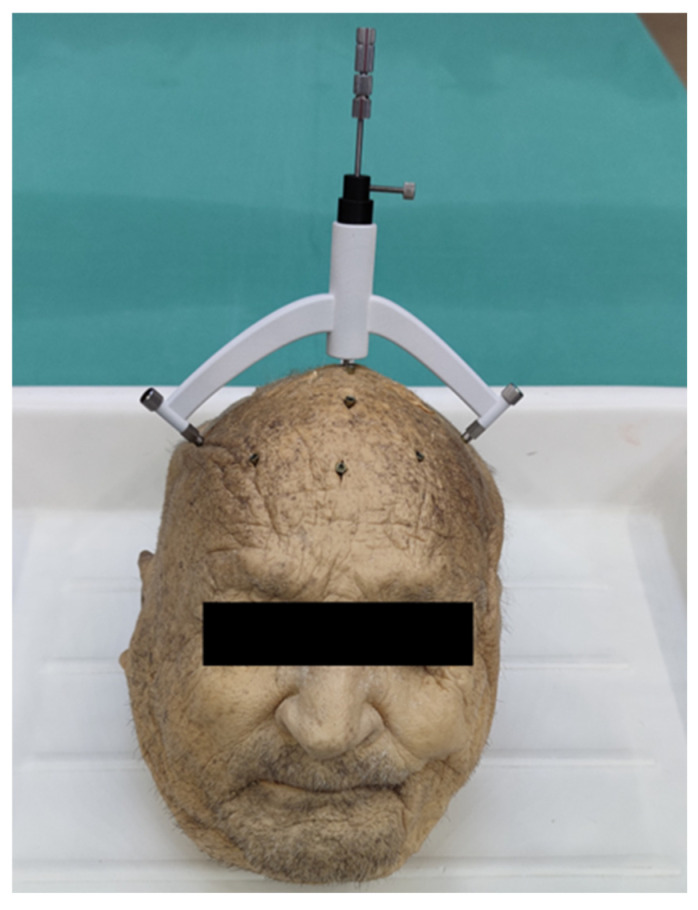
CT-controlled biopsy procedure by using patient-individual stereotactical biopsy device.

**Figure 5 bioengineering-11-01013-f005:**
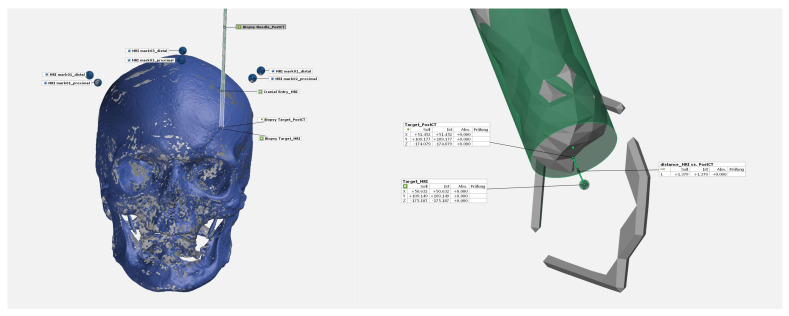
Synchronization of CT (pre and post) and MRI data for evaluation of target accuracy.

**Table 1 bioengineering-11-01013-t001:** Baseline data body donation.

Donor ID	Sex	Age	Height	Weight	Cause of Death
		[years]	[cm]	[kg]	
121-2019	male	89	171	77.0	Cardiovascular failure

**Table 2 bioengineering-11-01013-t002:** Results of stereotactic biopsy study.

Target Device ID (Brain Hemi-Sphere)	Device Weight [g]	Intracranial Lesion Depth [mm]	Segmented Needle Length [mm]	Segmented Needle Diameter [mm]	Deviation Planned vs. Real [mm]			
					x-Plane	y-Plane	z-Plane	Total
01 (right)	57.3	51.06	141.06	2.49	0.13	0.02	0.21	0.24
01 (left)		22.34	112.33	2.56	0.04	0.43	0.80	0.91
02 (right)	65.4	53.67	143.67	2.59	0.81	0.35	0.39	0.96
02 (left)		53.16	143.16	2.68	0.17	0.33	0.39	0.54
03 (right)	70.1	54.00	144.00	2.60	0.89	0.60	0.60	1.23
03 (left)		44.61	134.61	2.69	0.37	0.70	0.17	0.81
04 (right)	69.9	64.25	154.25	2.84	0.42	0.27	0.53	0.73
04 (left)		37.18	127.18	2.87	0.04	0.42	0.32	0.53
05 (right)	66.4	33.72	123.72	2.63	0.27	0.20	0.78	0.85
05 (left)		27.28	117.28	2.69	0.06	0.20	0.20	0.30
06 (right)	50.8	31.29	121.29	2.54	0.12	0.11	0.38	0.41
06 (left)		26.95	116.95	2.68	0.90	0.25	0.17	0.95
07 (right)	59.2	42.45	143.10	3.01	0.88	0.87	0.28	1.27
07 (left)		40.82	142.26	2.95	0.63	0.39	0.38	0.83
08 (right)	55.2	47.84	131.65	2.88	0.41	0.05	0.06	0.41
08 (left)		47.41	132.78	2.82	0.31	0.27	0.47	0.62
09 (right)	56.7	56.52	141.52	2.93	0.30	0.52	0.27	0.66
09 (left)		39.40	124.40	2.96	0.71	0.15	0.92	1.18
10 (right)	60.6	30.00	128.70	2.95	0.76	0.24	0.27	0.84
10 (left)		29.11	126.92	2.90	0.34	0.95	0.80	1.28
11 (right)	61.9	33.30	122.99	2.92	0.59	0.92	0.02	1.09
11 (left)		37.85	128.27	2.85	0.04	0.27	0.45	0.53
12 (right)	63.2	62.19	161.90	3.20	0.30	0.18	0.61	0.71
12 (left)		67.57	167.90	3.15	1.81	0.94	0.42	2.09
13 (right)	62.9	42.05	121.73	3.24	0.45	1.46	0.27	1.55
13 (left)		81.25	160.94	3.21	2.13	0.74	0.82	2.40
14 (right)	67.5	51.05	138.00	3.08	0.20	2.27	0.59	2.36
14 (left)		39.46	126.64	2.87	0.29	1.82	0.87	2.04
15 (right)	58.4	37.11	137.11	3.20	0.30	0.58	0.35	0.74
15 (left)		52.58	152.58	3.18	0.59	0.09	0.36	0.70
16 (right)	73.6	64.48	149.48	3.32	0.71	1.00	0.81	1.47
16 (left)		66.31	158.31	3.08	2.00	1.47	0.43	2.52
**mean_total_**	**62.4**	**45.88**	**136.77**	**2.89**	**0.56**	**0.60**	**0.45**	**1.05**
SD_total_	*6.0*	*13.86*	*14.19*	*0.23*	*0.53*	*0.53*	*0.24*	*0.63*
mean_right_		47.18	137.76	2.90	0.47	0.60	0.40	1.14
mean_left_		44.58	135.78	2.88	0.65	0.59	0.50	0.97

## Data Availability

The data that support the findings of this study are available from the corresponding author upon request.

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
