# Peer review of "ARISE—The Accuracy Evaluation of a Patient-Specific 3D-Printed Biopsy System Based on MRI Data: A Cadaveric Study"

_bioengineering, 2024, doi:10.3390/bioengineering11101013_

Round 1
Reviewer 1 Report
Comments and Suggestions for Authors
Not relevant to primary routine use and not applicable to daily use.
Only one head model was used, needle insertion should be under CT.
Probable CT-MRI mismatch could cause high risk bias.
Brain stimulation should be needed for this procedure.
Comments on the Quality of English Language
Minor editing
Author Response
We really appreciate your comments to improve our manuscript. Please find followed our description according to your questions.
Comment1:
Not relevant to primary routine use and not applicable to daily use.
Response1:
This type of individualization already represents a solution variant of stereotactic surgery in the Anglo-American region and is based on our own experience of such placement of stimulation electrodes in over 100 patients. This is representative of the fact that the technology of 3D-printed frame systems is associated with a high degree of accuracy and optimization of surgical procedures and operating time.
According to our own experience, this is associated with time savings, but not with a loss of precision.
We therefore also see a promising approach for brain biopsy using patient-specific 3D-printed systems.
Comment2:
Only one head model was used, needle insertion should be under CT. Response2:We evaluated 32 target points in different brain regions to consider a variability of bone anchor placement. In future studies more cadaver heads shall be applied for accuracy evaluation.
Our motivation is to perform a brain biopsy at the real patient without a CT and the hazard of x-ray.
Possible errors due to axis adjustment of conventional stereotactic systems are avoided, as all spatial axes are implemented in the 3D-printed system and no adjustment of distances and angles is required. This makes the stereotactic principle available to non-experienced neurosurgeons.
We applied the CT only as measurement instrument for the accuracy evaluation.
(This was changed in the manuscript line 312)
Comment3:
Probable CT-MRI mismatch could cause high risk bias Response3:With our approach only an MRI is required for image acquisition and trajectory planning. As we don’t need a CT we avoid errors due to image fusion. (line 293)
Comment4:
Brain stimulation should be needed for this procedure.
Response4:
After the application of our system for brain biopsy we also want to use it as further development for brain stimulation. (line 317)

Reviewer 2 Report
Comments and Suggestions for Authors
The article evaluates the initial experience with individually 3D printed stereotactic frames for brain biopsies. The biopsies were performed on cadaver head (32 biopsies) and the accuracy was evaluated with postoperative CT scans.
The introduction is well written and gives a good overview of the matter. In the materials and methods section, the authors correctly describe the procedure, how bone anchors were placed, 3D frames printed, biopsies performed and their accuracy evaluated. The tables and figures are informative enough. I only miss the information about the length of the procedures (how long does it take to make the 3D model, how long does the biopsy take). The authors should also describe more in detail how the 3D printing was performed and which factors were considered for its proper placement on the cadaver head. The results are well presented. In the discussion, the authors should include more studies that described the accuracy of brain biopsies and compare it with their results. They should also describe how their procedure would be performed on actual patients (for example where bone anchors would be placed, how many anchors would one actual procedure need etc.). They should also discuss the cost-effectiveness and length of their procedure compared to the standard use of stereotactic frame. In the limitations, they should focus on the future improvement of their procedure.
Overall, it is a novel and promising study where the authors achieved good initial diagnostic accuracy. The study should be considered for publication after proper corrections are made.
Comments on the Quality of English LanguageMinor editing needed
Author Response
We really appreciate your comments to improve our manuscript. Please find followed our description according to your questions.
Comment1:
The authors should also describe more in detail how the 3D printing was performed.
Response1:
The 16 devices and 32 length templates were manufactured from polyamide (type: PA 12) by using 3D printing technique MultiJetfusion (MJF 580 color (Hewlett-Packard Inc., Palo Alto, USA).
MJF technology is a powder bed process. PA12 in powder form is applied layer by layer in the building platform. After each individual layer, print heads are used to apply an agent in the areas where the 3D-model is to be produced. After this application, a thermal radiator moves over the current layer and heats it. The thermal energy is absorbed in particular by the powder particles sprayed with the agent, which then fuse together. This process is repeated for each additional layer.
After the printing process, the surface of the devices and templates were glass bead blasted with SMG 50 Rapid Ex machine (MHG GmbH, Düsseldorf, Germany).
(This was changed in the manuscript line 150)
Comment2:
Which factors were considered for its proper placement on the cadaver head. Response2:The calculated access corresponds to the shortest distance between the skull surface and the target point and defines the placement of the bone anchors. The trajectory is located inside the center of the placed bone anchors. (This was changed in the manuscript line 86)
Comment3:
The authors should include more studies that described the accuracy of brain biopsies and compare it with their results. They should also describe how their procedure would be performed on actual patients (for example where bone anchors would be placed, how many anchors would one actual procedure need etc.). They should also discuss the cost-effectiveness and length of their procedure compared to the standard use of stereotactic frame. In the limitations,They should focus on the future improvement of their procedure.
Response3:
Bjartmarz et al. compared the accuracy between conventional stereotactic and frameless systems for deep brain stimulation. Their results was 1.2 ± 0.6 for conventional and 2.5 ± 1.4 mm for frameless and systems.
The frameless VarioGuide and frame-based stereotactic system were analyzed by Bradac et al. with results of 2.7 ± 1.1 mm for the conventional systems and 2.9 ± 1.3 mm for the frameless VarioGuide.
Hodge et al. evaluated a frameless system for deep brain stimulation with an accuracy of 1.79 ± 1.02 mm.
In comparison to the mentioned studies our results with an accuracy of 1.05 ± 0.63 mm are comparable to conventional frame-based stereotactic systems.
The patient-specific 3D-printed stereotactic device eliminates all resterilization and reprocessing costs as well as maintenance, recalibration, repair and storage costs.
(This was changed in the manuscript line 270)
The operating time is approx. 2 hours for conventional systems. This time is also preceded by a CT scan under general anesthesia, trajectory determination, calculation and assembly of the stereotactic frame and axis adjustment.
This compares with an operating time of just 25 minutes with the new 3D-printed system.
(This was changed in the manuscript line 284)
Possible errors due to axis adjustment of conventional stereotactic systems are avoided, as all spatial axes are implemented in the 3D-printed system and no adjustment of distances and angles is required.
This makes the stereotactic principle available to non-experienced neurosurgeons.
(This was changed in the manuscript line 312)
This type of individualization already represents a solution variant of stereotactic surgery in the Anglo-American region and is based on our own experience of such placement of stimulation electrodes in over 100 patients. This is representative of the fact that the technology of 3D-printed frame systems is associated with a high degree of accuracy and optimization of surgical procedures and operating time.
According to our own experience, this is associated with time savings, but not with a loss of precision.
We therefore also see a promising approach for biopsies using patient-specific 3D-printed systems.

Round 2
Reviewer 2 Report
Comments and Suggestions for Authors
The manuscript has been sufficiently improved and can now be published in the journal
Comments on the Quality of English LanguageMinor editining needed